# Pro-Resolving Inflammatory Effects of a Marine Oil Enriched in Specialized Pro-Resolving Mediators (SPMs) Supplement and Its Implication in Patients with Post-COVID Syndrome (PCS)

**DOI:** 10.3390/biomedicines12102221

**Published:** 2024-09-29

**Authors:** Asun Gracia Aznar, Fernando Moreno Egea, Rafael Gracia Banzo, Rocio Gutierrez, Jose Miguel Rizo, Pilar Rodriguez-Ledo, Isabel Nerin, Pedro-Antonio Regidor

**Affiliations:** 1Sociedad Española de Médicos Generales y de Familia (SEMG), 28005 Madrid, Spain; asun@semg.es (A.G.A.); prodriguezl@semg.es (P.R.-L.); 2Solutex GC SL, Avenida de la Transición Espanola 24, 28108 Alcobendas, Spain; fmoreno@solutexcorp.com; 3Solutex GC SL, Parque Empresarial Utebo, Avda. Miguel Servet nº 81, 50180 Utebo, Spain; rafael.gracia@insercolab.com; 4OTC Chemo, Manuel Pombo Angulo 28-4th Floor, 28050 Madrid, Spain; rocio.gutierrez@chemogroup.com (R.G.); josemiguel.rizo@chemogroup.com (J.M.R.); 5Directora de la Cátedra SEMG-Estilos de Vida Unidad de Tabaquismo FMZ Profª Dpto. Medicina, Psiquiatría y Dermatología Facultad de Medicina, Universidad de Zaragoza, 50009 Zaragoza, Spain; isabelne@unizar.es; 6Exeltis Healthcare, Adalperostr. 84, 85737 Ismaning, Germany

**Keywords:** post-COVID syndrome, specialized pro-resolving mediators, inflammation, resolution

## Abstract

Objectives: This study aimed to evaluate the eicosanoid and pro-resolutive parameters in patients with Post-COVID Syndrome (PCS) during a 12-week supplementation with a marine oil enriched in specialized pro-resolving mediators (SPMs). Patient and methods: This study was conducted on 53 adult patients with PCS. The subjects included must have had a positive COVID-19 test (PCR, fast antigen test, or serologic test) and persistent symptoms related to COVID-19 at least 12 weeks before their enrolment in the study. The following parameters were evaluated: polyunsaturated fatty acids EPA, DHA, ARA, and DPA; specialized pro-resolving mediators (SPMs), 17-HDHA, 18-HEPE, 14-HDHA, resolvins, maresins, protectins, and lipoxins. The eicosanoids group included prostaglandins, thromboxanes, and leukotrienes. The development of the clinical symptoms of fatigue and dyspnea were evaluated using the Fatigue Severity Scale (FSS) and the Modified Medical Research Council (mMRC) Dyspnea Scale. Three groups with different intake amounts were evaluated (daily use of 500 mg, 1500 mg, and 3000 mg) and compared to a control group not using the product. Results: In the serum from patients with PCS, an increase in 17-HDHA, 18-HEPE, and 14-HDHA could be observed, and a decrease in the ratio between the pro-inflammatory and pro-resolutive lipid mediators was detected; both differences were significant (*p* < 0.05). There were no differences found between the three treatment groups. Fatigue and dyspnea showed a trend of improvement after supplementation in all groups. Conclusions: A clear enrichment in the serum of the three monohydroxylated SPMs could be observed at a dosage of 500 mg per day. Similarly, a clear improvement in fatigue and dyspnea was observed with this dosage.

## 1. Introduction

Emerging in China in November 2019, severe acute respiratory syndrome coronavirus 2 (SARS-CoV-2) provoked the first pandemic of the 21st century; COVID-19 (coronavirus disease 2019) spread worldwide within a few months and continues to impose an enormous burden on health systems and the economy. The transmission of the virions mainly occurs via droplet infection. Still, as they remain infective for up to 3 days, depending on the environmental conditions, they may also reach their hosts via everyday objects like computer keyboards, door handles, or furniture, finally entering the cells of the oral or nasal mucosa or via the conjunctiva of the eye [1].

The clinical manifestations of COVID-19 can vary, ranging from symptomless infections through intermediate courses of the disease to life-threatening manifestations with severe pneumonia, multiorgan failure, and death. Both mild and severe forms of COVID-19 may also lead to the so-called “Post-COVID Syndrome (PCS)”, a term representing the various symptoms caused by the disease that continue for months after the initial infection [2].

Globally, the disease mortality is about 3.4% [3], reaching up to 4.3% in Wuhan (China), where COVID-19 originated [4]. Comorbidities, mainly hypertension and diabetes, are directly linked to poor disease outcomes [5]. As no effective treatment options against COVID-19 have been developed, only symptomatic approaches are used in managing this disease; the antiviral drugs examined so far have not yet revealed convincing clinical efficacy [6] or still need extensive investigation, such as ivermectin [7]. While most (80%) symptomatic patients do not experience life-threatening manifestations of COVID-19, moderate disease courses can quickly become severe, leading to acute respiratory distress syndrome (ARDS) with multiorgan failure and death when no medical treatment occurs. Therefore, patients with moderate symptoms should also receive supportive treatment, including antiviral and/or antiphlogistic drugs, to prevent an aggravation of the disease. The treatment options under investigation include arbidol, chloroquine phosphate, ribavirin, favipiravir, ivermectin, interferon alpha-2b, and dexamethasone. Remedies with convalescent plasma or monoclonal antibodies like etesevimab and bamlanivimab have been investigated for their safety and efficacy in patients with COVID-19 [6,7]. However, as mentioned above, treatment is restricted to supportive and adjuvant care [8,9,10,11].

SARS-CoV-2 is comparable to the influenza virus in many respects, as both are RNA viruses that provoke respiratory symptoms ranging from very mild to highly severe forms that may result in a fatal course of the disease. Severe pathologies are often linked to overshooting reactions of the hosts’ immune system, mirrored by the so-called “cytokine storm”. This phenomenon is known for pathogens like the influenza virus or the Gram-negative bacterium *Francisella tularens*, which leads to pneumonia or hypercoagulation and is also observed in COVID-19 disease. Consequently, therapeutically suppressing the inflammatory immune response or the systemic use of active anticoagulants may represent promising approaches to managing COVID-19 symptoms [8].

Interestingly, it is not only ARDS that is associated with a poor outcome of the disease: in hospitalized patients with COVID-19, myocardial problems and kidney failure are observed to contribute to a fatal course of the disease [12,13,14]. However, the basic pathophysiological mechanisms of SARS-CoV-2 infections, which are responsible for damage to various tissue types, are not yet entirely understood [15], and coagulation processes are essential in this context [16,17]. Still, a systematic description of the underlying coagulatory and fibrinolytic processes and their relationship to the outcome of the disease has yet to be accomplished [18,19]. The exact mechanisms by which SARS-CoV-2 induces coagulatory and inflammatory responses and the interaction between these pathways during COVID-19 infection are therefore unclear.

Generally, the cessation and resolution of inflammatory processes depend on active strategies, which are largely driven by lipid mediator (LM) molecules, also called specialized pro-resolving mediators (SPMs) [20,21]. They are synthesized by cells of the innate immune system, which utilize the essential fatty acids arachidonic acid (ARA), eicosapentaenoic acid (EPA), n-3-docosapentaenoic acid (DPA), and docosahexaenoic acid (DHA) as substrates for enzymatic conversion to form four families of SPMs: lipoxins, resolvins, protectins, and maresins [22,23]. All SPMs are involved in actively regulating and enforcing the resolution of inflammatory processes, and due to their activity, for example, the amount of pro-inflammatory cytokines and chemokines is reduced at infected sites, and the influx of neutrophils is actively limited. Furthermore, macrophages are stimulated to enhance phagocytosis, kill bacteria, and perform the clearance of cell debris [20,22,23]. In animal disease models, an organ-protective action of SPMs has also been demonstrated [22]. Of particular interest in the present context is the observation that SPMs also seem to positively impact the alveolar fluid clearance (AFC) in ARDS, thereby supporting the physiological function reconstitution of the lung [24].

When lung tissue is injured, an immune response is triggered, which leads to an increase in the amount of pro-inflammatory molecules at the site of injury, followed by the entrance of immunocompetent cells into the alveolar space [25]. The influenza-A virus demonstrated a direct correlation between its virulence and the profound and continuous induction of the inflammatory response.

Its ability to disseminate into different tissues was not only associated with the strong activation of genes encoding for crucial elements of the pro-inflammatory cascade but it was also accompanied by a downregulation of the genes responsible for the lipoxin-mediated anti-inflammatory signaling pathways, thereby reducing the pro-resolutive capacity and protective role of the SPM [26]. Furthermore, for SARS-CoV-2 patients, a relationship between the lipid mediator profile and the severity of the disease has been demonstrated recently. Striking differences between the lipid profiles and abundance of certain LM derivatives were observed between severe and moderate courses of illness. A relationship between pre-existing comorbidities like BMI, diabetes, heart disease, and lipid profile changes has been described. It was speculated whether those risk factors led to a pre-existing imbalance in the LM profiles that might finally contribute to the severity of COVID-19 due to the decreased ability to counteract the inflammatory response induced by a SARS-CoV-19 infection [27].

In most cases, patients with COVID-19 feel better within a few days or weeks of the first symptoms’ appearance and fully recover within 12 weeks. However, for some people, symptoms can persist for weeks or months following the infection. The long-term effects of COVID-19 affect several body systems, including pulmonary, cardiovascular, and nervous systems, as well as psychological effects. These effects appear to occur irrespective of the initial severity of infection; even in mild or moderate cases, the disease can cause long-term organ damage but occurs more frequently in middle-aged women and those who initially show more symptoms (World Health Organization, 2021) [28].

Post-COVID Syndrome, also known as long-term COVID, occurs in individuals with a history of probable or confirmed SARS-CoV-2 infection, usually three months from the onset of COVID-19. Symptoms last at least two months and cannot be explained via an alternative diagnosis; they may appear following the initial recovery from an acute COVID-19 episode or the initial SARS-CoV-2 infection, and they may also fluctuate or relapse over time [28].

An exacerbated inflammatory response is recognized as a central component in many chronic diseases, including vascular diseases, metabolic syndromes, and neurologic diseases. The acute inflammatory response can be divided into two different processes, initiation and resolution, a process that was for many years considered passive [29]. Only after the discovery of the first mediators with pro-resolution capabilities did the processes leading to the resolution of the acute inflammatory response begin to be considered active [30,31]. The anti-inflammatory properties of omega-3 fatty acids have been known for a long time. These fatty acids compete with arachidonic acid, leading to lower levels of pro-inflammatory eicosanoids, and during the resolution process, omega-3 fatty acids produce signaling molecules such as resolvins, protectins, and lipoxins, specialized pro-resolution mediators known as SPMs. These SPMs are agonists that shorten the resolution of the inflammatory response via the stimulation of resolution key events, stopping the flow of neutrophils, improving the elimination of apoptotic cells, and causing bacterial death [32,33,34].

The food supplement investigated in this study is enriched in monohydroxylated SPMs, with previous studies showing it can increase SPM levels in serum and plasma in various physiological and pathological circumstances. During inflammation caused by a trauma or infection, there is a deficit of SPMs, and the administration of this new formula could significantly improve SPM levels in plasma and serum, as well as the ratio between SPMs and inflammatory prostaglandins.

Previous studies used high doses of EPA and DHA [35] or this new formula [36], and the common grounds for all the studies were as follows:The lack of adverse reactions;Significant rise in SPMs.

Considering the available data, the use of food supplements rich in omega-3 fatty acids will not be related to the onset of adverse reactions, and the expected rise in SPMs will be associated with a clinical improvement in the symptoms of patients with PCS, which, in turn, could endorse the use of the supplement as an addition for the management of the disease.

The measurement of the plasma and serum concentrations of pro-inflammatory (prostaglandins and leukotrienes) and pro-resolving lipid mediators (lipoxins, resolvins, protectins, maresins, and monohydroxylated mediators derived from EPA and DHA) in patients with PCS provided precious information about the immunological response of the patients regarding the inflammatory condition caused by the infection.

This study aimed at describing the immunological capacity and inflammatory response of this supplement on patients with PCS on the eicosanoid and pro-resolutive parameters and on the clinical entities dyspnea and fatigue compared to healthy individuals by establishing LM profiles and their precursor molecules in plasma and serum of the test groups and analyzing the clinical devolvement of the patients.

## 2. Material and Methods

This study was designed as a randomized, double-blind, placebo-controlled trial with four parallel supplement groups to assess the efficacy of a food supplement enriched in SPMs in patients with PCS. The measurements included the pro-inflammatory and pro-resolution lipid mediator levels and perceived fatigue and dyspnea measured through subjective questionnaires. The safety and tolerability of the investigational product (IP) were also evaluated.

This study was planned as a proof of concept, specifically a pilot study aiming to determine the effect of increasing food supplement doses. Two different amounts of the supplement were tested and controlled with a placebo. An additional low-dose group was added, independently of the other 2 and not regulated with the same objectives, to test the supplement’s effect on the levels of pro-inflammatory and pro-resolving lipid mediators.

Patients who were willing to participate signed the informed consent (IC) form, and those fulfilling all the inclusion criteria (and none of the exclusion criteria) were randomized to one of the four treatment options, three of which correspond to the double-blind placebo-controlled trial (A, B, C), and the fourth to the independent, non-controlled, low-dose group (X). No follow-up phase was planned for after this study.

The following procedures were performed during each visit of the study:

Screening visit—V0 (day 7/day 3):Conduct an anamnesis and physical exam;Measure the body temperature, blood pressure, and heart rate;Encourage the patient to participate in the study;Give oral and written information and obtain informed consent;Check the inclusion/exclusion criteria;Review the current concomitant medication.

Randomization visit—V1 (day 1):

This visit took place between 3 and 7 days after the screening visit.

Check the inclusion/exclusion criteria;Conduct a physical exam;Measure the body temperature, blood pressure, and heart rate;Take a blood sample;Perform a pregnancy test (if applicable);Perform a Fatigue Severity Scale (FSS) test;Perform a Modified Medical Research Council (mMRC) validated Dyspnea Scale test;Performed randomization;Review the record of adverse events (AEs);Assess the concomitant medication;Review the record of intercurrent or concomitant illness;Provide the IP(s);Provide the patient’s diary;Provide instructions about the completion of the diary.

Interim visit—V2 (day 28 ± 3):

Four weeks after the beginning of the treatment (±3 days), the patients returned to the center where they attended the interim visit.

Perform physical exam;Measure the body temperature, blood pressure, and heart rate;Take a blood sample;Perform a Fatigue Severity Scale (FSS) test;Perform a Modified Medical Research Council (mMRC) Dyspnea Scale test;Review the record of adverse events (AEs);Assess the Concomitant medication;Review the record of intercurrent or concomitant illness;Return the empty and unused product containers;Review the patient’s diary;Provide the IP(s).

End of Study visit—V3 (day 84 ± 3):

Twelve weeks after the first administration of the IP, the patients returned to the same center for the final visit.

Perform a physical exam;Measure the body temperature, blood pressure, and heart rate;Take a blood sample;Perform a Fatigue Severity Scale (FSS) test;Perform a Modified Medical Research Council (mMRC) Dyspnea Scale test;Review the record of adverse events (AEs);Assess the concomitant medication;Review the record of intercurrent or concomitant illness;Return the empty and unused product containers;Review the patient’s diary.

### 2.1. Study Population

The study was conducted in 53 adult patients with PCS. The subjects included must have had a positive COVID-19 test (PCR, fast antigen test, or serologic test) and persistent symptoms related to COVID-19 at least 12 weeks before their enrolment in the study. The candidates were selected directly by their respective treating centers, which informed the potential candidates about the survey and offered them participation in the trial.

No study procedure was conducted before the patient gave written consent, including their signature, name, and surname. The investigation team member providing the study information also had to sign the informed consent sheet.

The trial protocol was designed and conducted following the ethical principles defined in the Declaration of Helsinki, and all procedures were consistent with GCP and the applicable regulatory rules. Table 1 depicts the demographic data and sex distribution of the enrolled patients.

Patient characteristics and inclusion and exclusion criteria

To be included in the study, the participants must meet all the following inclusion criteria:

Inclusion criteria:(1)Adult patients with Post-COVID Syndrome, both genders, between 18 and 70 years old.Patients with clinical criteria that prove the COVID-19 infection: Diagnosis confirmed using a COVID-19 test (PCR, rapid antigen test, serological test). Symptoms must persist longer than 12 weeks after the beginning of the symptoms.Patients with fatigue/asthenia, dyspnea, and one of the following conditions:General malaise;Headaches;Low mood;Muscular pain.(2)Body mass index between 18.5 and 30 kg/m^2^.(3)The ability to provide informed consent.(4)Women who participate in the study must comply with one of the following conditions:Unable to become pregnant: women who had had surgical sterilization or were over two years after menopause.Fertile women must have a negative pregnancy test prior to their inclusion in the study (conducted during screening) and be using a highly efficient contraceptive method: hormonal contraceptives, intrauterine devices, condoms together with spermicide and gel, partner’s surgical sterilization (vasectomy), or total sexual abstinence during the study. The use of these contraceptive methods must continue at least 3 months after the last dose of the study products.

Exclusion criteria:

To participate in the study, patients must meet none of the following exclusion criteria:(1)Pregnant or breastfeeding women.(2)Inability to use a highly efficient contraceptive method.(3)Recruited in another clinical trial.(4)Subjects involved in another clinical trial 4 weeks prior to their inclusion.(5)Patients with any concomitant illness or condition that could significantly affect the hematologic, renal, endocrine, pulmonary hepatic, gastrointestinal, cardiovascular, immunologic, central nervous, dermatologic, or any other system, with the exceptions stated in the inclusion criteria.(6)Use of immunosuppressant drugs or prolonged or maintained use of anti-inflammatory drugs and/or corticoids.(7)Hypersensitivity, allergy, or idiosyncratic reaction to omega-3 acids, fish or soya allergies.

Removal of Patients from Therapy or Assessment

Subjects were free to withdraw from the study at any time. The investigator could withdraw a subject from the study due to the onset of adverse events, safety concerns, or protocol non-compliance, which could have jeopardized the validity of the data. A thorough objective and subjective monitoring of the status of each patient, their symptoms, and their adherence to the study procedures was thus conducted during the scheduled visits.

### 2.2. Supplement Allocation

The investigational product (IP) was a food supplement enriched in SPMs, formulated as capsules. Each soft gel contained 500 mg of a marine lipid fraction, standardized to 17-HDHA, 14-HDHA, and 18-HEPE. Both the investigational product and the placebo were manufactured, packed, and labeled by Laboratorios Liconsa SL; Spain.

The IP is obtained from fish body oil of wild-caught anchovies [*Engraulis ringens* and/or *Engraulis encrasicolus* and/or *Anchoa nasus*] and/or sardines [*Sardinops sagax sagax* and/or *Sardina pilchardus* and/or *Sardinella longiceps*] and/or mackerel [*Scomber scombrus* and/or *Scomber colias* and/or *Scomber japonicus* and/or *Trachurus murphyi*] from the Pacific and Atlantic oceans.

This study’s sponsor provided all the investigational products adequately masked, except for group X, who were not blinded, for whom the products were thus not masked.

Three centers participated in the study.

The study lasted 12 weeks, including a baseline visit and three study visits.

The treatment allocation was performed by randomly assigning each subject to a treatment group or the placebo group. The randomization ratio was 3:3:1:3 [16/16/5/16]):Group A: N = 16 patients;Group B: N = 16 patients;Group C: Placebo N = 5 patients;Group X: N = 16 patients.

The dosage was as follows:Group A = 3000 mg/day;Group B = 1500 mg/day;Group C = Placebo;Group X = 500 mg/day.

### 2.3. Ethical Approval

The ethical committee approved the investigational trial and centers: Comité de Ética de la Investigación Santiago-Lugo with the number 2012/097.

Clinical Trial Registry: ISRCTN13270662

### 2.4. Primary Endpoint

Blood samples from patients with PCS.

Analytical Procedure

Blood samples were drawn over three days, each treated as a mono-replicate. They were separated into plasma and serum, subjected to standard preparation procedures, and stored at −80 °C until further analytical processing. All samples were analyzed individually, and the results were used for statistical analysis (see the relevant chapter below).

Extraction and profiling of lipids and lipid mediators via LC-MS/MS

In this study, lipid mediator laboratory analyses were conducted at Solutex GC SL. In brief, the extraction of lipid mediators from plasma and serum samples involved a solid-phase extraction (SPE) process. Plasma or serum samples were mixed with internally deuterium-labeled standard solutions at 500 pg, enabling the quantification of analytes. After protein removal through precipitation and centrifugation, SPE was performed using established protocols. Following elution from the SPE column using organic solvents, extracts were dried and resuspended before injection into an LC-MS/MS system.

The LC-MS/MS system was utilized with a binary eluent system. The elution gradient program and flow rate were carefully controlled. The negative ionization mode and scheduled Multiple Reaction Monitoring (MRM) acquisition were used for analysis. Quantification was achieved by calculating the area under the peaks, and identification was achieved by employing an MS/MS library to match signature ion fragments for each molecule, while retention times for each lipid mediator were compared to internal standards. The study ensured the optimization of lipid mediator parameters for accurate quantification.

### 2.5. Statistical Analysis

Arithmetic means, standard error, and minimum and maximum values were calculated and displayed for each patient and analyte. The software package GraphPad Prism version 9.0.2 (San Diego, CA, USA) was used for outlier exclusion with default parameters ROUT (Q = 1%).

A ratio between pro-inflammatory and pro-resolutive parameters was calculated to establish a measure for the balance between the pro-inflammatory and pro-resolutive axes of the underlying physiological processes.

Quantitative variables were described as their average ±SD or 95% percent confidence intervals. Qualitative variables were described as frequencies and percentages. Changes from the baseline were calculated using the ANOVA test, using multiple testing corrections (or the non-parametrical equivalent if the variable does not follow a normal distribution).

*p*-values below 0.05 were rated statistically significant as there was no adjustment for multiple testing. The data presented here are merely explorative and descriptive.

Analyzed lipids and lipids mediators

The following analytes were determined:

Polyunsaturated fatty acids: EPA, DHA, ARA, DPA.

Monohydroxylated SPMs: 17-HDHA, 18-HEPE, 14-HDHA.

SPMs: resolvins (RvE1, RvD1, RvD2, RvD3, RvD4, RvD5), maresins (MaR1, MaR2), protectins (PD1, PDX), lipoxins (LXA4, LXB4).

Pro-inflammatory eicosanoid lipid mediators: prostaglandins (PGE2, PGD2, PGF2α.), thromboxanes (TxB2), leukotrienes (LTB4).

### 2.6. Secondary Endpoint

As a secondary efficacy objective, the evolution of the above-mentioned parameters until the fourth week of treatment (day 28) was calculated.

Further secondary efficacy variables are as follows:Fatigue Severity Scale (FSS) test: The FSS test measures fatigue on a unidimensional scale. It consists of nine questions with seven possible answers, quantifying each item on a scale of 1 to 7. The evolution of the mean scores from baseline to visit 2 (4th week of treatment, day 28) and to the end of the study (day 84 of treatment) is calculated.Modified Medical Research Council (mMRC) Dyspnea Scale test: The scale includes 5 degrees of physical activity that could cause dyspnea. The scale punctuates the dyspnea from 0 (no exercise causes dyspnea) to 4 (the dyspnea prevents the patients from leaving the house or performing routine daily activities like dressing up). The baseline results are compared to the scores at visit 2 (day 28) and the end of the study (day 84).

Safety

To assess the safety of the IP, all adverse events (AEs) that occurred to the participants during the study, since the first administration of the IP up to the last visit (treatment-emergent AEs), were collected, assessed, and recorded in the CRF, regardless of their relationship to the study product. The events that would have begun before the supplementation were included in the subject’s clinical history.

The AEs could be clinically significant abnormalities found in the vital signs (body temperature, heart rate, or blood pressure) or during the physical exam, or they could be reported directly by the subjects to the investigators either during their visits or in their diaries. The investigators had to record the AEs in the CRF and assess their intensity, seriousness, and causal relationship with the IP using their best medical judgment and experience.

## 3. Results

Laboratory changes

In this observational study, we observe that the quantification of each parameter was detectable in the sera but not in the same way in the participants’ plasma.

This study aimed to quantify the targeted SPM eicosanoid lipidomics in human plasma and serum profiles. We measured ARA, DHA, and the EPA metabolome using the “state-of-the-art” targeted LC-MS/MS metabololipidomics after the intake of three different dosages of the marine oil-enriched solution containing the daily dose of 500 mg, 1500 mg, or 3000 mg.

After quantification, the summation of the total derived pro-resolutive mediators resulted in a statistically significant increase (*p* < 0.05) when comparing each of the three metabolites (14-HDHA + 17-HDHA + 18-HEPE [ng/mL]) in the serum of the patients at all dosages.

### 3.1. Values for 14-HDHA

14-HDHA [ng/mL]: Serum concentrations of 14-HDHA ranged from 0 to 300 ng/mL in most cases (treatments per week), with a few values above these numbers. The increase could be seen throughout the 12 weeks of use. Figure 1 depicts these values. The difference between starting the treatment (baseline before intake of the product) and the end of treatment was significant in all groups (*p*-value = 0.002) but not between the groups.

### 3.2. Values for 17-HDHA

17-HDHA [ng/mL]: In most cases, serum 17-HDHA concentrations ranged from 0 to 100 ng/mL.

The difference between starting the treatment (baseline before intake of the product) and the end of the treatment was significant in all groups (*p*-value = 0.0007) but not between the groups.

Figure 2 depicts the value distribution for 17-HDHA.

### 3.3. Values of 18-HEPE

18-HEPE [ng/mL]: Serum concentrations of 18-HEPE ranged from 0 to 50 ng/mL in most cases (treatments per week). The difference between starting the treatment (baseline before intake of the product) and the end of the treatment was significant in all groups (*p*-value = 0.00001) but not between the groups.

Figure 3 depicts the values of 18-HEPE.

### 3.4. Total Amount of the Three Monohydroxylates

During the supplementation, there was a significant increase in the sum of all three parameters. This shows the efficacy of the supplementation. The differences before and after the supplementation were significant. Figure 4 depicts the data (*p*-value: 0.0005).

### 3.5. Sum of Pro-Inflammatory Values

The sum of pro-inflammatory markers was calculated by adding the values of PGE2 + PGD2 + PGF2α + TXB2 + LTB4 [pg/mL] in each register:

The cumulative sum of all measured pro-inflammatory markers did not change significantly during the supplementation; there was a slightly reducing trend in supplementation, *p*-value = 0.232. Figure 5 depicts the development in all three groups.

### 3.6. Ratio between Pro-Inflammatory and Pro-Resolutive Markers

Pro-inflammatory/monohydroxylated ratio: The ratio between pro-inflammatory (sum) and monohydroxylated (sum) markers was calculated for each record.

There was a significant change in the ratio. During the supplementation, an improvement in the ratio in all three groups could be observed, showing the high efficacy of the supplementation.

The ratios decreased after the first four weeks and continued to decline until the end of the 12 weeks of supplementation. Figure 6 depicts the ratio.

These changes were significant, with a *p*-value of 0.025.

### 3.7. Clinical Changes

To determine the effects of the investigated product on the clinical manifestations of long-term COVID, as secondary objectives of the study, the impact of the IP on the patients’ fatigue and dyspnea, two of the most prevalent symptoms observed in these patients, was assessed. The secondary efficacy variables are as follows:Changes in the Fatigue Severity Scale (FSS) scores from baseline until weeks 4 and 12;Changes from baseline until weeks 4 and 12 in the mMRC (Modified Medical Research Council) Dyspnea Scale score.

Both scales are commonly used and validated methods to assess either fatigue or the degree of functional disability due to dyspnea.

The evolution of these clinical variables, including the four treatment groups, was analyzed using a mixed general linear model.

### 3.8. Fatigue

The differences between the baseline FSS scores and 4 and 12 weeks after treatment were calculated. All groups tend to improve the fatigue symptoms included in the FSS questionnaire. No significant differences are detected among the four treatment groups, but a clear trend in improvement can be seen in group X, which used 500 mg of marine oil per day.

Figure 7 depicts the data:

An improvement of 9% in the mean value for fatigue could be observed between weeks 4 and 12 for the patients on a dosage of 500 mg daily.

### 3.9. Dyspnea

The differences between the baseline and weeks 4 and 12 in the mMRC scale scores were calculated for each patient. The Chi-squared test was used for the analysis of the differences between treatments. For the differences between baseline and week 12, X-squared = 8.2496 and *p*-value = 0.509; between baseline and week 4, X-squared = 7.3615 and *p*-value = 0.600. A slight improvement can be observed for each group regarding the frequency and percentage of patients in each grade of the scale. However, there were no significant differences in the evolution of the mMRC scores among the four treatment groups during the study. Figure 8 shows the development of the mMRC scores for each treatment group at baseline (1), after four weeks of treatment (2), and at the end of the study, after 12 weeks of treatment (3). The data reveal an overall slight improvement in the mMRC scale in all groups at the end of the study, whereby most of the patients included experienced no or 1 point of improvement. The analysis revealed no differences among the study groups (see Table 2 and Table 3).

## 4. Discussion

This study observed a significant difference in the amount of the lipid mediators 14-HDHA, 17-HDHA, and 18-HEPE after supplementing with a marine oil-enriched formulation in patients with Post-COVID Syndrome.

Given that there were no statistically significant differences between the dosages of 500 mg, 1500 mg, and 3000 mg, a clear trend of favoring a 500 mg dosage per day can be postulated.

SARS-CoV-2 and PCS can lead to a robust inflammatory response, represented by a high abundance of pro-inflammatory signaling molecules like interleukin-6 and C-reactive protein, an increased erythrocyte sedimentation rate, and increased fibrinogen levels [37,38,39,40].

It was demonstrated through the evaluation of LC-MS/MS data that the ratio between (pro-inflammatory) eicosanoid derivatives and pro-resolutive lipid mediator molecules was significantly improved by using the lipid mediators.

Previous studies showed that pro-inflammatory markers were in higher abundance in SARS-CoV-2-affected subjects than in healthy ones [41]. In a recent study, the eicosanoid and pro-resolutive parameters of patients with COVID-19 with severe symptoms, such as ARDS, were compared to the lipid profiles of only moderately affected patients, the results showing that the lipid mediator products of ALOX12 and COX2 decreased, while those of ALOX5 and cytochrome P450 increased [27].

The pro-thrombotic alterations observed in patients with COVID-19 may derive from processes initiated by the damage of virus-infected cells. In patients who require intensive care treatment, high levels of pro-inflammatory cytokines were detected compared to subjects with a moderate manifestation of the disease [17]. Those extensive inflammatory processes may lead to aggravated coagulatory reactions.

An important diagnostic parameter for the induction of coagulation is the increase in D-Dimer levels, and for COVID-19, it has become an indicator of the severity of the disease. Subjects who develop DIC (disseminated intravascular coagulopathy) or sepsis have a high mortality risk [42,43,44]. The processes leading to these severe coagulopathies are not yet entirely understood. However, the underlying inflammation gives rise to coagulatory alterations instead of the virus.

On the other hand, hemorrhagic bleeding disorders are not observed in the context of SARS-CoV-2 infections, which contrasts with other single-stranded RNA viruses, such as Ebola [3]. In addition, data from Wuhan support the conception that the inflammatory host response leads to coagulopathies via interlinked signaling pathways.

A comprehensive cohort study demonstrated a relationship between activated neutrophils, platelets, and the dysregulated coagulation cascade that finally led to immunothrombotic damage in various tissues. Utilizing coagulation tests with peripheral blood samples and histopathological analyses, the authors identified the systemic hypercoagulability with microvascular thrombosis observed in several organs as characteristic key contributors to severe manifestations of ARDS in COVID-19. Consequently, platelet and neutrophil counts and signs of coagulation cascade activation were suggested as valuable pharmaceutical targets for the treatment of COVID-19 [45]. Therefore, the systematic surveillance of coagulation processes combined with prophylactic anticoagulant therapy has become essential for managing COVID-19-affected patients.

The pure elimination of the infectious agent may not be sufficient to re-establish homeostasis in affected patients. Still, a relatively active cessation of inflammatory processes and clearing of infection sites is required.

It has been demonstrated in mouse models that thrombi were markedly reduced when the factor resolvin D4 (RvD4) was applied. The treatment also led to decreased neutrophil infiltration and a higher abundance of monocytes in a pro-resolutive state and cells in the early stages of apoptosis. RvD4 also triggered the enhanced biosynthesis of further pro-resolutive resolvins of the D-series family. The SPMs, mainly RvD4, were shown to be important modulators of the gravity of thrombo-inflammatory processes while furthering the resolution of thrombi [46].

Interestingly, in patients with coronary arterial disease, certain pro-resolutive SPMs are reduced compared to healthy subjects. However, when treated with pharmacological doses of EPA and DHA for one year, a clear shift in the lipid mediator profile compared to non-treated patients was observed with a decrease in triglyceride levels and pro-inflammatory prostaglandins and a significant increase in certain pro-resolutive SPMs. The SPM-triggered macrophage-based phagocytosis of clots was enhanced in patients treated with the SPM precursors [35].

The first studies of this nutritional supplement have demonstrated its effectiveness in raising SPMs in plasma in different physiological and pathological conditions.

Having detected a significant deficit of SPMs in conditions of inflammation, and as described in the protocol, it is estimated that applying this new formulation will substantially improve both the SPMs in plasma and serum and the ratio between SPMs and prostaglandins.

In one study [36], it was possible to see that the ideal doses of DHA, EPA, and monohydroxylates lay between 1500 mg and 3000 mg. The findings common to all studies were as follows:(a)Zero incidence of side effects;(b)Substantial increase in SPMs.

### Limitations of the Clinical Trial

The limitations of this pivotal trial include the small number of recruited patients, which also did not allow us to segment between men, women, and age.

Another limitation may be the fact that there is a natural trend of improvement in symptoms in patients with Post-COVID syndrome.

To further shed light on the role of SPMs in COVID-19 disease, it will be informative to validate our data in clinical trials. This supplementation might be beneficial by preventing the cytokine storm observed in severe manifestations of COVID-19 disease, as the SPMs may enforce the pro-resolutive axis of inflammatory processes. This also helps improve chronic courses associated with heart and lung tissue inflammation. In addition, supplementation with SPMs or their precursor metabolites may improve pathologic conditions for recovered or vaccinated subjects. As demonstrated in this study, the increase in SPMs observed in the sera of patients with PCS might be effective in managing this chronic situation.

Furthermore, the improvement in fatigue and dyspnea is promising. Supplementation with this marine oil enriched in SPMs hence represents an approach to managing patients with PCS.

Clinical Trial Registry: ISRCTN13270662

## 5. Conclusions

A clear enrichment in serum of the three monohydroxylated SPMs could be observed also at a dosage of 500 mg per day. In the same way, a clear improvement in fatigue and dyspnea was observed with this dosage.

In conclusion, the use of selective pro-resolving mediators, including monohydroxylates, holds promise for the management of various acute and chronic diseases across a wide range of medical conditions. Particularly in lung obstructive diseases, supplementation with enriched marine oil nutritional products shows potential for attenuating serious complications such as dyspnea and fatigue. However, further research is necessary to determine the optimal dosing regimens and fully elucidate the mechanisms underlying their therapeutic effects.

This is also important, as we could not find any correlations between the different dosages and the development of the analyzed serum parameters.

Nonetheless, this study’s findings suggest that these interventions may represent a valuable approach to addressing inflammatory diseases and mitigating obstetrical complications, thereby improving health outcomes.

## Figures and Tables

**Figure 1 biomedicines-12-02221-f001:**
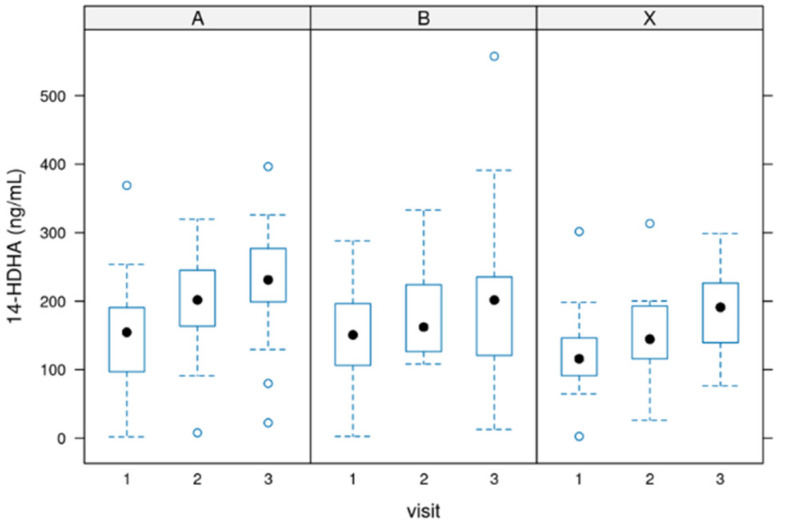
Distribution of the 14-HDHA values during the 12 weeks of the study. Group A = 3000 mg per day, Group B = 1500 mg per day and group X = 500 mg per day of dosage. Circle: outlayres. Black dot = median value.

**Figure 2 biomedicines-12-02221-f002:**
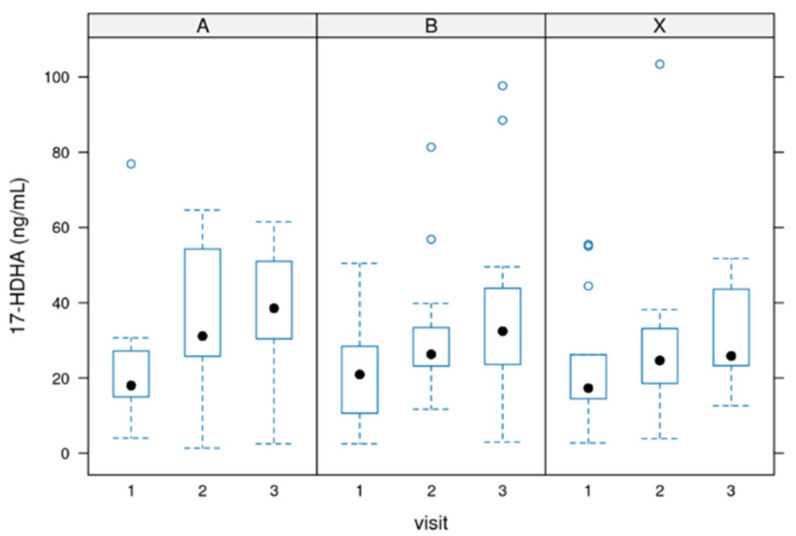
Distribution of the 17-HDHA values during the 12 weeks of the study. Group A = 3000 mg per day, Group B = 1500 mg per day and group X = 500 mg per day of dosage. Circle: outlayres. Black dot = median value.

**Figure 3 biomedicines-12-02221-f003:**
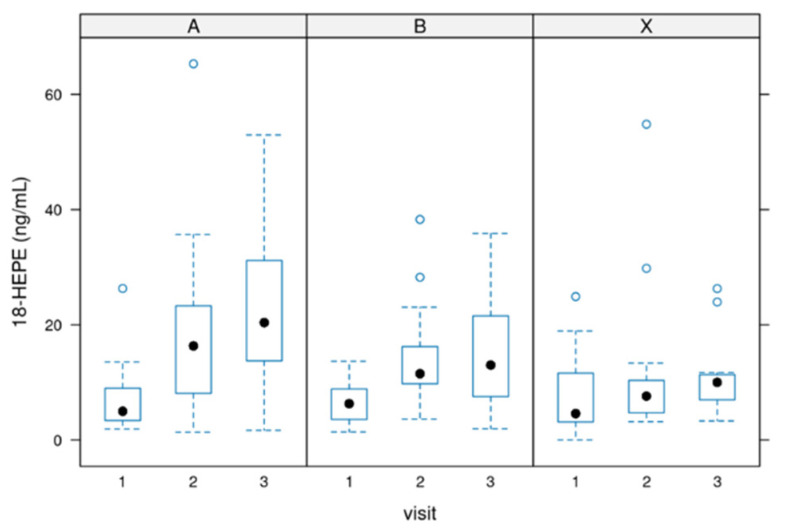
Distribution of the serum values for 18-HEPE. Group A = 3000 mg per day, Group B = 1500 mg per day and group X = 500 mg per day of dosage. Circle: outlayres. Black dot = median value.

**Figure 4 biomedicines-12-02221-f004:**
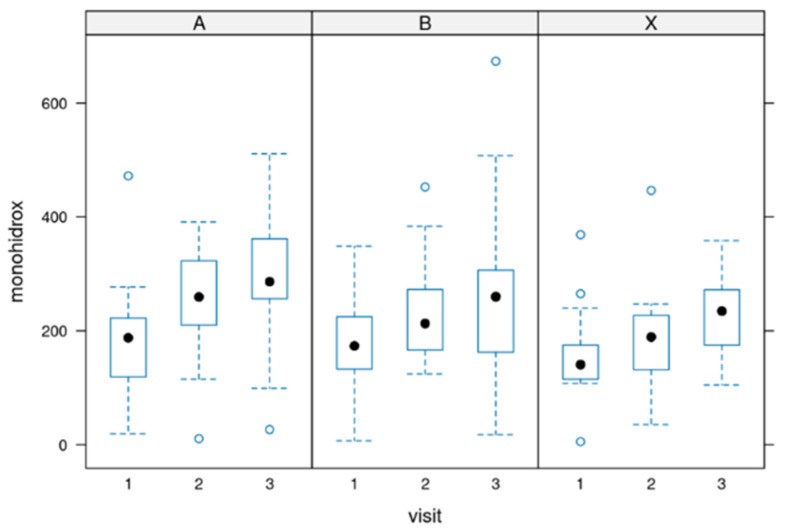
Sum of the three investigated monohydroxylated parameters. During the treatment there was a significant increase of 14-HDHA, 17-HDHA and 18-HEPE in all groups. Group A = 3000 mg per day, Group B = 1500 mg per day and group X = 500 mg per day of dosage. Circle: outlayres. Black dot = median value.

**Figure 5 biomedicines-12-02221-f005:**
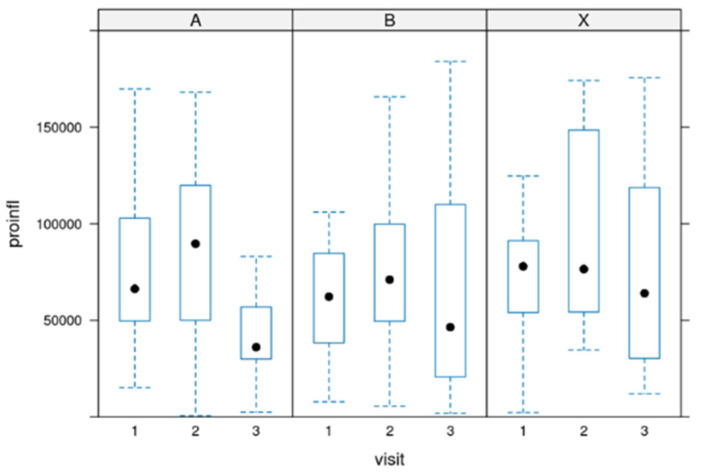
Sum of the pro-inflammatory parameters. Group A = 3000 mg per day, Group B = 1500 mg per day and group X = 500 mg per day of dosage. Circle: outlayres. Black dot = median value.

**Figure 6 biomedicines-12-02221-f006:**
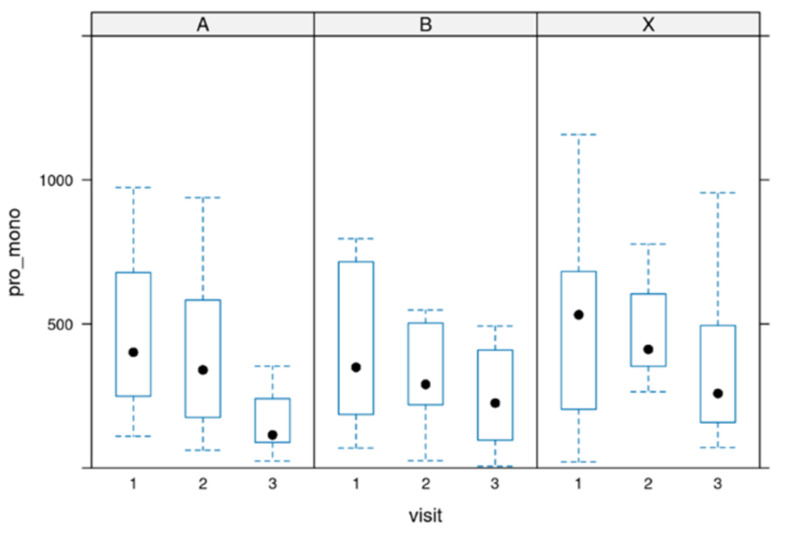
Development of the ratio between inflammatory and resolutive parameters. The differences were significant. Group A = 3000 mg per day, Group B = 1500 mg per day and group X = 500 mg per day of dosage. Circle: outlayres. Black dot = median value.

**Figure 7 biomedicines-12-02221-f007:**
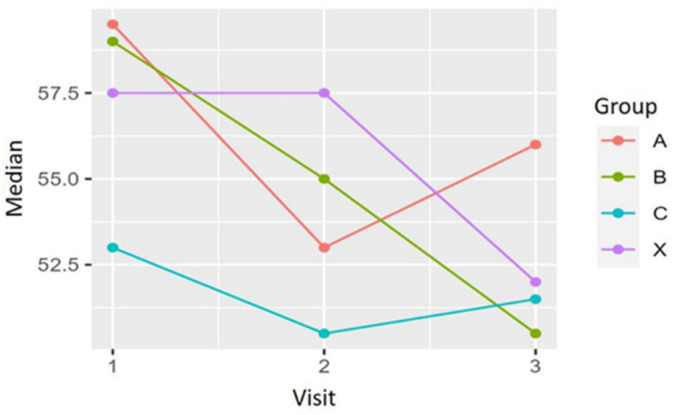
Development of the fatigue scale for the three treatment groups. A clear trend to a clinical improvement can be observed. The group A, B and X were the ones receiving the active substance in the dosage of 3000 mg, 1500 mg and 500 mg per day respectively.

**Figure 8 biomedicines-12-02221-f008:**
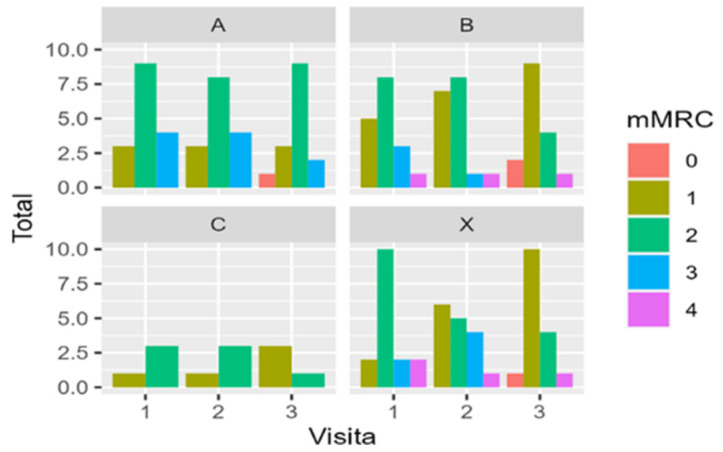
Development of the dyspnea scale for the three treatment groups. A clear trend to a clinical improvement can be observed. Group A = 3000 mg per day, Group B = 1500 mg per day, Group C = Placebo and group X = 500 mg per day of dosage.

**Table 1 biomedicines-12-02221-t001:** The following chart depicts the study chronogram:.

Assessment	Screening Visit	Randomization Visit	Interim Visit	EoS Visit
	V0	V1	V2	V3/FDE
	Day 0 (−3 to −7 Days)	Day 1	Day 28 (±3 Days)	Day 84 (±3 Days)
Informed consent	X			
Inclusion/Exclusion criteria	X	X		
Randomization		X		
Medical history	X			
Vital signs (T^a^, blood pressure, heart rate)	X	X	X	X
Physical examination	X	X	X	X
Blood sample extraction		X	X	X
Pregnancy test		X		
Fatigue Severity Scale (FSS)		X	X	X
Modified Dyspnea Scale (mMRC)		X	X	X
Adverse events		X	X	X
Concomitant medication	X	X	X	X
Concomitant diseases	X	X	X	X
Delivery of the study product		X	X	
Delivery of the patient’s diary		X		
Product accountability			X	X
Review of adherence to the dosing schedule			X	X
Review of the patient’s diary			X	X
EoS = End of Study				

**Table 2 biomedicines-12-02221-t002:** Numerical data of the ananlysed SPMs, pro inflammatory factors and the ratio between both for different time periods during the clinical trial.

Proinflammatory (=PRO) SUM: PGE2 + PGD2 + PGF2α + TXB2 + LTB4 [pg/mL]											
SERUM											
Week	0	4	12		0	4	12		0	4	12
Mean	76,176	88,550	67,412		64,135	79,864	67,223		162,425	159,746	51,720
SPMs in ng/mL											
W0–W4											
SERUM											
	500 mg			SERUM	1500 mg			SERUM	3000 mg		
SPMs	167.33	201.91		SPMs	169.94	225.44		SPMs	184.44	250.07	
Ratio PRO/ SPMs	455.2562	438.5612	4%	Ratio PRO/SPMs	377.4012	354.2561	6%	Ratio PRO/SPMs	880.6602	638.7988	27%
W4–W12											
SERUM											
	500 mg			SERUM	1500 mg			SERUM	3000 mg		
SPMs	201.91	214.84		SPMs	225.44	278.06		SPMs	250.07	290.90	
Ratio PRO/SPMs	438.5612	313.7754	28%	Ratio PRO/SPMs	354.2561	241.7588	32%	Ratio PRO/SPMs	638.7988	177.7974	72%
W0–W12											
SERUM											
	500 mg			SERUM	1500 mg			SERUM	3000 mg		
SPMs	167.33	214.84		SPMs	169.94	278.06		SPMs	184.44	290.90	
Ratio PRO/SPMs	455.2562	412.1652	9%	Ratio PRO/SPMs	377.4012	287.2231	24%	Ratio PRO/SPMs	880.6602	549.1507	38%

**Table 3 biomedicines-12-02221-t003:** Single values of dyspnea during the 12-week treatment.

Changes between Week 12 and Baseline Number and % of Patients that Have Experienced Changes in mMRC Score: −2, −1, 0 or 1					
Treatment	−2	−1	0	1	Total
A	0 (0)	5 (33.33)	10 (66.67)	0 (0)	15 (100)
B	2 (12.5)	6 (37.5)	7 (43.75)	1 (6.25)	16 (100)
C	0 (0)	2 (50)	2 (50)	0 (0)	4 (100)
X	3 (18.75)	8 (50)	5 (31.25)	0 (0)	16 (100)
Data are displayed as N (%). Chi-square: X-squared = 8.2496, df = 9, *p*-value = 0.509					

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
