# Peer review of "Pro-Resolving Inflammatory Effects of a Marine Oil Enriched in Specialized Pro-Resolving Mediators (SPMs) Supplement and Its Implication in Patients with Post-COVID Syndrome (PCS)"

_biomedicines, 2024, doi:10.3390/biomedicines12102221_

Round 1

Reviewer 1 Report

Comments and Suggestions for Authors

The study evaluated eicosanoid and pro-resolutive parameters in 53 Post-COVID Syndrome (PCS) patients over 12 weeks of supplementation with marine oil enriched in Specialized Pro-resolving Mediators (SPMs). The parameters included various polyunsaturated fatty acids and eicosanoids, and clinical symptoms like fatigue and dyspnoea were assessed. Significant increases in SPMs (17-HDHA, 18-HEPE, 14-HDHA) and a decrease in the pro-inflammatory to pro-resolutive lipid mediator ratio were observed. Supplementation improved fatigue and dyspnoea, with noticeable benefits at even the lowest dosage of 500 mg per day

Major concerns about the study:

1.      The study lacks some very basic and crucial information like not disclosing the exact source and nature of the marine oil supplementation makes it impossible to replicate the study and reproduce the same results.

2.      The manuscript does not tell anything about the inclusion and exclusion criteria for the patients such as age, gender, and other disease history of the patients which makes the results less valuable.

3.      No blood lipid profiling of the individuals participating in the study was carried out at the start of the treatment, so how can it be concluded that the changes in the lipidome were due to the marine oil supplementation?  

4.      The number of participants in each group is too low to draw significant conclusions in this type of study model.

5.      Titles of the figures are there but the figure legends are missing.

6.      There should be a separate conclusion section in the MS

Minor concerns about the manuscript:

1.      The numbering of headings and subheadings of different sections is missing in the manuscript.

2.      The same terminology should be used consistently throughout the manuscript to avoid confusion. For instance, the spelling of dyspnoea/dyspnea and mMRC/MMRC is different in different places.

3.      The standard units and symbols should be used uniformly throughout the manuscript such there should be space between “-80” and “-80” in Line 302, add space between “500” and “pg” in line 311, there should be no space between “9” and “%” in line 450.  

4.      There are numerous typographic/spelling mistakes such as “inmonohydroxylated” instead of “in monohydroxylated” in line 162. Change “Patient” to “Patients” in line 27, “centreto” in 231 to “centre to”, replace “+” with “,” in line 386, and “evolution” to “evaluation” in line 439, etc. Similarly, many common names have been unnecessarily started with capital letters.

5.      Elaborate on all the abbreviations used in the manuscript.

Regards 

Comments on the Quality of English Language

The English language is mostly fine, however, the manuscripts need careful revision for typographic mistakes. 

Author Response

The point to point replay is attached 

Reviewer 2 Report

Comments and Suggestions for Authors

The manuscript entitled “Pro-resolving inflammatory effects of a marine oil enriched in SPMs supplement and its implication in patients with a Post Covid Syndrome (PCS)” represents a double-blind placebo-controlled study, which examines the effect of marine supplement enriched with polyunsaturated fatty acids on the level of a number of lipids, dyspnea and fatique parameters in a small sample of patients with post-COVID19 syndrome.

Several issues need to be addressed.

1.     This study lacks a description of possible limitations, including that placebo group is rather small (N = 5), as well as a sample in total. Please, add such subsection at the end of the manuscript.

2.      Since the sample size is rather small, it is required to calculate the correspondence to the normal distribution of quantitative outcomes and use corresponding statistical criteria for the analysis. It is unlikely that data are normally distributed; hence, the authors have to use a non-parametric criterion instead of Student’s t-test.

3.     Corresponding t-test values (or I suggest to add the values for non-parametric tests) are absent in the Results.  The Results section should include distinct values of statistical parameters, i.e. each statistically significant difference has to be reported in brackets after each finding. It remains unclear whether any differences in SPMs, FSS, mMRC were examined between the groups (i.e. A, B, X vs placebo), please, report corresponding statistical values. In turn, although the manuscript has many figures, it would benefit from adding more tables even if the differences between the groups remain statistically non-significant. For instance, the authors should include the table on the differences in the level of examined lipids in four groups at different time points.

4.     Please, check the correctness of Figure 1 y-axis. In addition, I recommend to rename the axes in the figures, i.e. to give a complete explanation of axes with reporting units of measurement for better understanding. In turn, figures require adding of explanations in captions. In addition, please add some marks on figures to clarify statistically significant differences between the groups (i.e. brackets).

5.     How can the authors explain a decrease in B and X groups at time point “3” and an increase in group A for the same time point? I suggest discussing this issue.

6.     I suggest changing the term “lipidome” throughout the article, since the present study examines the effect of marine supplement on the level of certain lipids rather than the whole lipidome.

 I would suggest to accept after major revision, providing that the authors addressed all the comments.

Author Response

See document

Reviewer 3 Report

Comments and Suggestions for Authors

The article evaluated the implication of the Pro-resolving inflammatory effects of a marine oil enriched in SPM supplement in patients with a Post Covid Syndrome (PCS). The results support the conclusion, however, the following concerns should be addressed.

1. The introduction section is very long, please reformat it to focus it related to the manuscript. The text related to SPMs may be in a separate section or the first 12 paragraphs should be shortened.

2. The inclusion and exclusion criteria are not mentioned clearly (all patients with COVID-19 with 2 weeks of symptoms is the only inclusion criteria mentioned). Please include the age, sex, comorbid conditions, treatment taken, phase of COVID-19, (mild, moderate, severe, or asymptomatic), recovered from which disease status, what the symptoms/organ system involved during PCS, etc

3. What was the duration these patients were enrolled, were they from a single center or multiple center?

4. Protocol approval number?

5. The results for pro-inflammatory and resolvins (SPMs) are not clear (Figure 9). Inclusion of the supplementary data with the levels of various SPMs will support Figure 9. The data for the levels of resolvins should be included.

6. Resolvins affect inflammatory cell recruitment and cytokines levels- were the levels of these were investigated?

7. Please include the limitations of the study.

8.  In the discussion section, various aspects like inflammation and coagulopathy (with related manifestations) have been discussed, it will be better to focus on one aspect either inflammation or coagulopathy or general symptoms. Further, text discussed for RvD4 in mouse model and like that has been discussed which seems nothing to do with the results if the data for RvD4 is not available or significantly changed with treatment. Please focus on the results in the discussion.

Author Response

See document

Round 2

Reviewer 1 Report

Comments and Suggestions for Authors

Thank you for modifying the manuscript. My only concern remains is the high similarity, especially 22% with one internet source alone. 

Author Response

Thansk 

We have tried to reduce this to a maximum.

Reviewer 2 Report

Comments and Suggestions for Authors

The authors have addressed all my previous comments except for the following one:

I suggest changing the term “lipidome” throughout the article, since the present study examines the effect of marine supplement on the level of certain lipids rather than the whole lipidome.

Author Response

We have changed the 2 words of lipidome that were in the text into : eicosanoid and pro-resolutive parameters

Reviewer 3 Report

Comments and Suggestions for Authors

Thank you for revising the manuscript. In figures 1-6, the standard deviation is more than the mean/average and the positive StdDev is different than the negative StdDev. Please explain. The analysis should be checked by a statistician.

Author Response

Thansk for the comment.

The SD is alwasy from the point and they are equal.